# In Vivo Application of Silica-Derived Inks for Bone Tissue Engineering: A 10-Year Systematic Review

**DOI:** 10.3390/bioengineering9080388

**Published:** 2022-08-15

**Authors:** Nicolas Touya, Ayako Washio, Chiaki Kitamura, Adrien Naveau, Yasuhiko Tabata, Raphaël Devillard, Olivia Kérourédan

**Affiliations:** 1Univ. Bordeaux, INSERM, BIOTIS, U1026, F-33000 Bordeaux, France; 2Division of Endodontics and Restorative Dentistry, Department of Oral Function, Kyushu Dental University, 2-6-1 Manazuru, Kokurakita-ku, Kitakyushu 803-8580, Japan; 3Faculty of Dentistry, University of Bordeaux, 146 rue Léo Saignat, F-33076 Bordeaux, France; 4CHU de Bordeaux, Pôle de Médecine et Chirurgie Bucco-Dentaire, Place Amélie Raba Léon, F-33076 Bordeaux, France; 5Laboratory of Biomaterials, Department of Regeneration Science and Engineering, Institute for Frontier Life and Medical Sciences, Kyoto University, Kyoto 606-8397, Japan

**Keywords:** biofabrication, bioprinting, ink, tissue engineering, bone regeneration, 3D printing, silicon, silica, silicium, in vivo

## Abstract

As the need for efficient, sustainable, customizable, handy and affordable substitute materials for bone repair is critical, this systematic review aimed to assess the use and outcomes of silica-derived inks to promote in vivo bone regeneration. An algorithmic selection of articles was performed following the PRISMA guidelines and PICO method. After the initial selection, 51 articles were included. Silicon in ink formulations was mostly found to be in either the native material, but associated with a secondary role, or to be a crucial additive element used to dope an existing material. The inks and materials presented here were essentially extrusion-based 3D-printed (80%), and, overall, the most investigated animal model was the rabbit (65%) with a femoral defect (51%). Quality (ARRIVE 2.0) and risk of bias (SYRCLE) assessments outlined that although a large majority of ARRIVE items were “reported”, most risks of bias were left “unclear” due to a lack of precise information. Almost all studies, despite a broad range of strategies and formulations, reported their silica-derived material to improve bone regeneration. The rising number of publications over the past few years highlights Si as a leverage element for bone tissue engineering to closely consider in the future.

## 1. Introduction

Bone defects, whether due to infection, tumor, trauma, surgery, aging or genetic disorders, are a major health issue, affecting hundreds of millions of people every year [1]. Although a human bone graft remains the gold standard therapy, the demand outmatches the supply. A wide range of synthetic biomaterials has been developed to overcome limitations associated with the use of autografts and allografts such as morbidity on the donor site for autografts, and the lack of availability and risks of disease transmission, infection and immunogenicity for allografts [2]. Xenografts and xenogenic products are an uprising therapeutic field but also tend to suffer from graft drawbacks. Synthetic materials offer a more affordable alternative; however, they can be criticized because of their limited osteoconductivity, inadequate mechanical properties, unpredictable dissolution kinetics and poor reproduction of bone architecture [3]. Hence, the optimum formulation to overcome these hurdles is still to be found. Moreover, the development of tissue engineering has led to the design of new materials and strategies aiming at bone tissue regeneration instead of bone tissue replacement [4].

During the last decades, silica-based materials have gained a rising potential in hard tissue engineering since their first application in 1971 [5,6] and have been widely used for medical applications [7].

Silicon (Si) is the eighth-most common element in the universe by mass. More than 90% of the Earth’s crust is composed of silicate minerals, making silicon the second most abundant element in the crust (about 28%) after oxygen [8]. Many studies have shown the important biological role of Si in humans [9] and revealed its presence at various rates in many tissues within the body, such as the liver, kidney, lungs, muscle, bone, cartilage and other connective tissues [10].

At the bone level, silicon is an essential nutrient for tissue formation. In the early 1970s, a deficiency of Si was demonstrated to be involved in bone formation disorders [11,12]. Many authors reported enhanced bioactivity of biomaterials due to the presence of Si, which promotes collagen type I synthesis, osteoblastic differentiation and bone repair [13]. Indeed, Si plays a key role in connective tissue metabolism, particularly in bone and cartilage [14]. In osteoporosis and osteopenia, the reduced proliferation and activity of osteoblasts were found to be correlated to the loss of the biological availability of Si [15]. Other experimental studies have shown the role of Si in the remodeling process of bone and suggested its effect on the inhibition of the physiological resorption mechanisms [16,17].

These findings highlight the obvious interest and high potential of silicon inclusion in biomaterials used for bone regeneration.

Easily available, and suitable for malleable preparations, silicon-derived materials crossed the path of additive manufacturing [18]. Developed in the second part of the 20th century [19], the application of this field to biology covers various technologies, used for fabricating 3D objects in mineral and/or organic elements [20,21]. In bone tissue engineering, sustaining the required mechanical functions of bone is one of the historical and critical challenges [22]. As such, material composition must be compatible with strategies of construct modeling, such as extrusion-based bioprinting, the most common approach [23]. Another crucial point is to induce a favorable biological response to restore or supply the lost bone function. Often, the desired outcome is to locally obtain the mineralization of the tissue replacing the bone defect, which will eventually be remodeled as a bone tissue as mature and physiologically competent as native bone. Therefore, research is heavily focused on developing suitable materials that bring a satisfying answer on these two levels, exhibiting robust mechanical properties with a high biocompatibility and strong regenerative potential. Thus, in the scope of identifying the most efficient materials for bone tissue engineering, investigations of silicon-derived materials were extensively conducted, as it has been widely demonstrated to be an essential component of different types of biomaterials such as bioactive ceramics in bone repair/regeneration [24].

In bone regeneration and repair applications, silicon has often been used as a substitute or a dopant in calcium phosphate-based ceramics since it has a positive impact on the performance of conventional biomaterials, related to the surface chemical structure, mechanical strength, bioactivity and biocompatibility [25]. Moreover, the current trend has shifted from traditional biocompatible materials to bioactive materials, to promote a specific biological response between the material and the tissue, leading to the development of a bond between them [26].

Indeed, several research studies have been conducted in the past [27,28], notably, regarding physiologically relevant compounds such as hydroxyapatite [29] or collagen [30], but none specifically concerned with the use of silicon.

Of the numerous materials and strategies assayed, some printable silicon-containing formulations went through the in vitro characterization to evaluate their in vivo bone repair potential. Thus, this systematic review proposes to report on the last decade of in vivo applications based on (bio)printed silicon strategies for bone tissue engineering.

## 2. Materials and Methods

### 2.1. Protocol

The systematic review was conducted according to the Preferred Reporting Items for Systematic Reviews and Meta-Analysis (PRISMA) guidelines [31], using the Population, Intervention, Comparison and Outcome (PICO) methods [32] to define the search strategy. The protocol was registered in PROSPERO database (CRD42022311293).

### 2.2. Focused Question

The following focused question was defined: “Which silica-based ink formulations have been shown effective for bone tissue regeneration?”

### 2.3. Selection Criteria

In vivo preclinical studies involving the use of silica-based ink for bone regeneration were included. Only studies published in English with their abstracts available on the database were considered. In vitro and ex vivo studies, case reports and reviews were excluded. 

### 2.4. Search Strategy

The search strategy was developed based on the PICO reporting system (Table 1). A structured electronic search for articles published in English was conducted up to and including 10 April 2022 using the electronic databases MEDLINE (PubMed), Scopus and Web of Science. 

The following search combination was used: 

PubMed: ((silic*[Title/Abstract]) AND (print*[Title/Abstract]) AND (bone[Title/Abstract])) NOT (silicone[Title/Abstract]) AND ((in vivo[Title/Abstract]) 

Scopus: (TITLE-ABS-KEY (silic* AND print* AND bone AND NOT silicone) AND TITLE-ABS-KEY (in AND vivo))

Web of Science: **silic*** (Topic) AND **print*** (Topic) AND **bone** (Topic) NOT **silicone** (Topic) AND **in vivo** (Topic)

The keyword “silicone” was intentionally removed from the research outcome, being already described as a poor, detrimental material for bone repair [33]. Thus, all silicone-based approaches were excluded from the research scope.

Additional articles complemented the electronic search after manually screening the list of references of all publications selected. The searches were re-run just before the final analyses to retrieve the most recent studies eligible for inclusion. 

### 2.5. Screening Methods and Data Extraction 

All references were imported into a Zotero database. After eliminating the duplicated ones, references were exported to an Excel sheet to perform the screening phases. The articles were selected by two reviewers (NT and OK) working independently. The selection was performed in two phases, based on inclusion and exclusion criteria: (1) title/abstract screening, (2) full-text screening. The two lists of selected references were then compared and all disagreements were resolved by discussion, or, if persistent, by a third reviewer (RD).

Prioritized exclusion criteria were as follows: (1) during title/abstract screening, no application in bone regeneration, reviews, in vitro or ex vivo studies, clinical trials, cases reports, no silica-based material ink, (2) during full-text screening, no relevant outcome measure reported, no control group.

For eligible studies that matched with inclusion criteria, full text was screened to extract data by two reviewers using the data extraction form and was summarized in the form of Excel-based extraction tables. Data were extracted from text, tables and graphs. Extraction information included but was not limited to: data for evidence synthesis (study design, animal model, intervention of interest, primary and secondary outcomes), as well as information required for assessment of study quality and assessment of risk of bias. In case of missing data, the corresponding authors were contacted via email. Any disagreement between reviewers was resolved by discussion with the third reviewer (RD).

### 2.6. Science Mapping Analysis

Science mapping analysis of scientific domains was performed from the list of included articles by using keyword co-occurrence networking on VOSviewer (free software, version 1.6.18, Centre for Science and Technology Studies, Leiden University, Leiden, The Netherlands, 2022). Network analysis of the keywords was generated from the matrix of retrieved papers [34] (threshold value at 4). The terms–document matrix allowed to measure document similarities between clusters of topics.

### 2.7. Quality Assessment and Risk of Bias

The methodological quality and risk of bias of the studies included were assessed by two independent reviewers.

The ARRIVE 2.0 guidelines for reporting animal research were used to evaluate the quality of the studies. The items evaluated according to ARRIVE 2.0 [35] (“reported”, “not reported”, “unclear”) were: (1) study design, (2) sample size, (3) inclusion and exclusion criteria, (4) randomization, (5) blinding, (6) outcome measures, (7) statistical methods, (8) experimental animal, (9) experimental procedures, (10) results, (11) abstract, (12) background, (13) objectives, (14) ethical statement, (15) housing and husbandry, (16) animal care and monitoring, (17) interpretation/scientific implications, (18) generalizability/translation, (19) protocol registration, (20) data access, (21) declaration of interests.

The risk of bias was evaluated using the SYRCLE (Systematic review Center for Laboratory Animal Experimentation) tool for animal studies [36], using the following criteria: (1) random sequence generation, (2) baseline characteristics, (3) allocation concealment, (4) random housing, (5) blinding of participants and personnel, (6) random outcome assessment, (7) blinding of outcome assessment, (8) incomplete outcome data, (9) selective outcome reporting, (10) other risk of bias. Three categories of potential bias judgments were used: “low risk of bias”, “high risk of bias”, “unclear risk of bias”.

Two reviewers (NT and OK) were involved in assessment of risk of bias and study quality. When no consensus was reached, discrepancies were resolved by a third reviewer (RD).

### 2.8. Data Analysis

Data analysis was performed in a descriptive way due to heterogeneity of the included studies in terms of animal model, bone defect model, interventions and outcomes. No meta-analysis was considered for this systematic review. Summary tables were intended to provide the most relevant information on study characteristics and findings. In particular, the descriptive analysis explored the relationships between studies and described the impact of various silica-based ink formulations on bone regeneration process.

## 3. Results

### 3.1. Systematic Review following PRISMA Guidelines

After application of the inclusion and exclusion criteria, 175 articles were selected initially. 

The search using the PICO model, yielded 43 articles from MEDLINE/PubMed, 62 articles from Scopus and 70 articles from Web Of Science within the past 10 years. After the removal of duplicates/triplicates and preliminary analysis of the abstracts, 95 articles were selected for full-text analysis and evaluation, after which 44 articles were excluded and 51 articles were included in the review (Figure 1).

The spatial representation of the relationships between the keywords was displayed through a science map (Figure 2). The core, most recurrent keywords were (by decreasing rank): “tissue scaffold”, “animals”, “osteogenesis”, “printing, three-dimensional” and “bone regeneration”. 

This occurrence representation map highlighted three silicon forms used within the scoped articles: silicon, silicon dioxide and silicates. With one form excluding the others, those keywords had no link strength between each other, and were located in opposite areas. Of the material formulation-related keywords drawn by the occurrence map, silicon was observed to be closest to ceramics, while silicon dioxide was associated to calcium phosphate and silicates was near polyesters and calcium compounds.

The most occurring physical characteristics were porosity, compressive strength and X-ray diffraction. “Mesenchymal stem cells” was among the top 30 most recurring words/phrases and the only cellular population mentioned, and the most represented biological analyses were neovascularization, cell proliferation and differentiation.

### 3.2. Study Characteristics

In the scoped articles, three animal models were assessed (Figure 3): rabbit (65%), rat (29%) and mouse (6%). Interestingly, no large animals (sheep, goat) were involved. It is worthy to note that the number of articles was here correlated to the size of the model. Two preferential bone models were used: femur (51%) and calvaria (25%). In this review, only bone regeneration was considered as a crucial criterion, and no specific consideration was given to mechanical assessment. However, except calvaria and maxillary, all bone models in this review are associated with motion and superior load-bearing capacity.

Seven assembling strategies were performed in the articles gathered for this review. An overwhelming number of studies employed extrusion-related techniques (80%), and the six other strategies shared the other 20%. Within this minority, inkjet-associated strategies were the most representative [37,38,39] (6% of total), while digital light processing [40,41] and manual deposit or injection [42,43] were both 4%. Electrospinning [44], laser-assisted bioprinting [45] and selective laser melting [46] shared equally the 6% remaining (2% each). The trend of the studies assessing silicon-based inks was here to obtain an extrudable paste, mostly mixed with polyvinyl alcohol (PVA), that went through a sintering process after the printing step.

To summarize, the most recurrent combination of models and approaches assessed in the present review were: rabbit–femur–extrusion. The probability of retrieving a study (among the scoped articles) investigating the extrusion of a scaffold implanted in a femoral defect of rabbit was calculated and found to be P_(e×f×r)_ = ((0.8 × 0.51 × 0.65) = 0.2652 (26.25%)). In this study, P_(e×f×r)_ × N_(articles)_ = 13.52 articles should therefore theoretically associate those three designs. Seventeen articles (33.3% of total) were actually found to present those characteristics, giving a 6.8% difference between the predicted value and the actual number of articles assessed. 

The main data from the articles selected are shown in Table 2 and Figure 4, and additive information may be found in the Appendix A.

The list of articles was divided into two main categories, for silicon was either part of the native material (1), or was added as an external component of the initial material assessed (2). In each category, two additional subcategories were created to sort the studies depending on the fact that the silicon has either a primary role (a), or a secondary role (b) over the bone regeneration assessed (Table 3). 

By creating these four categories, all the articles retrieved via the algorithm employed were able to be sorted without the possibility of an article belonging to more than one category. Closest articles were then gathered by focus of interest, creating final subcategories with the largest size possible.

Three main sources of Si were identified: tetraethyl orthosilicate (TEOS) (37%), pure SiO_2_ (29%) and wollastonite (CaSiO_3_) (20%). The remaining 14% of sources were shared by Ca_2_Si, Ca_3_Si, Na_2_SiO_3_ and RSiO_3/2_, or were of undefined origin. According to the sorted articles, TEOS and CaSiO₃ were principally involved as scaffold basal constituents, while SiO_2_ free forms were mostly presented as doping agents.

### 3.3. Outcomes

#### 3.3.1. Articles’ Sorting and Main Outcomes

Category 1—Si is in the native material

With 78.4% of the total studied articles sorted in this category, the use of Si as a native part of the investigated material was the leading category.

-
*1a—Si is in the native material with a primary role*


Within articles including Si as a native part of the material, 21.6% of the total (11 out of 51) articles placed silicon as a primary element of focus. The compositions of the investigated materials were variable with monocalcium-based inks [47,48], tricalcium-based inks [45,55]; glasses also involved [53,54]; elements such as magnesium [28,29,30], strontium [52] or lithium [56] were also core components of calcium–silicate materials. Controls were usually either an empty defect, β-TCP or titanium scaffolds. Only Touya et al. [45], using laser-assisted bioprinting in a mouse calvaria defect model, reported an absence of enhanced tissue mineralization despite encouraging in vitro results, whether associated with cells or not. Overall, the outcome was the improvement in bone repair compared to the control.

-
*1b—Si is in the native material with a secondary role*


The largest subcategory, including 56.8% of the total (29 out of 51) articles reviewed, could be divided into three main focuses: doping an existing scaffold, assessing a new material formula (in which Si is of minor interest for the authors) or investigating physical differences between scaffolds with the same composition. Scaffolds were doped by either organic, non-organic or mixed elements with coating or loading strategies.

Two articles implanted materials loaded with cells. Shen et al. [62] reported no significant difference in bone formation between loading or not their magnesium and calcium–silicate scaffold with bone marrow stromal cells. Wang et al. [74] used gingival fibroblast and observed an improved bone regeneration compared to their sole implants.

Plant-derived organic compounds [58,59,65], animal growth factors [60,63,64,73], osteopontin-derived peptide [61] and decellularized matrix [57] were investigated. Overall, they all triggered an enhanced bone regeneration compared to their respective non-loaded controls.

Most non-organic doping involved magnesium [67,69,70,71,72], while others included copper [44], strontium [66] or graphene [68]. All of the studies reported an improvement in bone regeneration associated with loaded materials compared to their controls.

In several studies [43,75,76,77,78,79,80], Si was part of the native material, but not considered as a determining factor in the bone repair evaluated. Although Meseguer et al. [78] only performed qualitative observations, all studies reported a bone repair enhancement.

For three studies, physical properties of the material assessed constituted the focus of the investigations: porosity [40,41] and surface texture [81] were highlighted to be determining parameters for bone repair response.

This category illustrates the use of Si as a structural/vehicle agent.

Category 2—Si is an additive material

Among the scoped articles, 21.6% of the total were sorted in this category.

-
*2a—Si is an additive material with a primary role*


Si was used as a determining additive material in 19.6% of total studied articles.

All but one of the articles in this category reported the use of Si as a doping agent [37,38,39,42,82,83,84,85,86] meant to be released through time in the implant micro-environment. Zhang et al. [46] presented the coating of a titanium scaffold with a bioglass. All articles reported an improved bone regeneration compared to their siliconless counterparts (for Zhang et al., at the periphery of the implant).

-
*2b—Si is an additive material with a secondary role*


Only one (2% of total) article was sorted in this category [87], as the element of focus was a tri-molecule dexamethasone-FGF2-phenamil loading in a chitosan/bioglass-coated PCL scaffold. Silicon was associated to the coated bioglass on the PCL scaffold, and was one ingredient out of a vast association of doping agents. This study also reported an improved bone regeneration.

#### 3.3.2. Quality Assessment and Risk of Bias

Results from the quality assessment of the studies selected for detailed analysis are shown in Figure 5.

All scoped articles were analyzed according to SYRCLE and ARRIVE 2.0 methods. As ARRIVE is, to date, not associated with any graphical representation, the performed quality assessment was depicted inspired by the SYRCLE display tool robvis [88].

According to SYRCLE evaluation, 86.5% of answers to each risk of bias were “Unclear” as the information needed was either missing or incomplete for most studies, while 11.5% were attributed as “Low” when information was unequivocal and 2% as “High”. If one or more item of any article was filled by a “High” potential risk of bias, the whole article was systematically considered to be also of “High” potential risk of bias, regardless of the element that led to the decision to attribute the initial “High”. Two categories were identified as most involved in sorting High risks of bias: the animal random housing (cat. 4) and the treatment of outcome data (cat. 8–9). Randomness of housing could not be ensured as it was not necessarily specified in many studies. Some studies presented differences between the number of animals used displayed in the methods section, and the number of animals displayed in the results section without justifying it.

With the ARRIVE 2.0 method, 75% of items were “Reported”, 25% were “Unclear” and only 1% were found to be “Not reported”; the categories with most missing or partially reported data were the inclusion and exclusion criteria, and the blinding.

Overall, the articles were found to be of satisfying quality but associated with risks of bias. However, those risks could be mostly attributed to a lack of information provided by authors rather than clearly misconducted experiments.

## 4. Discussion

Silica-based materials exist in various forms, depending on the purity of sole Si, or complexation, resulting in a hybrid material. Even though Si was found in the literature to be part of natural metabolism and to benefit from a high renal clearance [89], some forms of silica may exert toxicity or deleterious effects at the scale of the whole organism [90,91]. None of the articles in this review seemed to employ such Si derivatives. Investigating the possibility to use Si as a drug delivery carrier, the size of 70 nm and below [92] of the Si particles applied to skin was found to be responsible for systemic exposure in mice and exerted mutagenic activity [93]. Even though such findings were consistent with the inhalation or digestion of such a size of nanoparticles [94,95], the literature insists on the good compatibility of these sizes with the dermal route even at a 2000 mg/kg dose [96]. In this review, the Si nanoparticles’ smallest reported form was of 100 nm diameter [84], and no systemic effect was reported. However, Si content within the ink formulations investigated was unassessable due to the ink preparation process. Although the initial source and proportion of Si or Si-based derivatives within the material was indicated in many studies, the washing, drying and sintering steps contributed to affect the material mass or volume, and therefore the element’s contents. Moreover, in different articles, the purpose of Si was to be released through time. Hence, measures of the native material’s Si content regarding this strategy were irrelevant, and the focus on the outcome, i.e., the bone repair compared to appropriate controls, constituted the principal focus of this review.

Indeed, Si, beyond being cytocompatible, was reported to exert a bioactive effect, by enhancing angiogenesis in endothelial cells [97], and enhancing mineralization [98] in osteogenic lineages. These two critical events being tightly associated to an efficient bone repair, the direct impact of Si upon them was investigated.

However, other articles excluded from our scope used alternative strategies. For instance, in a study by Li et al. [99] (rejected for assessing only the subcutaneous implantation of material), the strategy relied on the support of activated and polarized macrophages to stimulate angiogenesis consecutive to bone defect creation. Mediated by Si ions, M2 macrophages were polarized. Associated with HUVEC, they significantly improved angiogenesis and enhanced bone repair subcutaneously. The immune system could therefore be a first order actor to modulate the tissue repair response after a trauma, of which silicon may offer a therapeutic interest.

With the emergence of biofabrication strategies for tissue engineering, the development of compatible materials on both technological and therapeutic aspects has increased, especially recently [23]. Indeed, more than half of the recovered articles through the algorithm were published in 2019 or later, highlighting the uprising place of material inks developed in bone tissue engineering. Among the initially gathered articles, 51 studies remained in the scope. The article sorting was designed in the perspective of best answering the original question “Which silica-based ink formulations have been shown effective for bone tissue regeneration?”. Categories were designed to avoid duplicates, and no article was found to belong to multiple subcategories. This sorting could help decipher relevant trends of the use of Si in recent Si-based materials.

Indeed, it was outlined that in ~60% of articles, the role of silicon was not the determining aspect of the study. Overall, Si was considered as holding a structural role within printed materials. For the doped materials, when Si was often brought alongside with Zn, Mg or other elements, it was not possible to formally consider Si as the unique discriminated agent of bone repair enhancement, except for two studies. A trend of 96% of studies nevertheless reported a bone repair improvement in the formulations involving Si.

This review focused on 3D printing-associated materials. Although the printing of silicium is not new, its application essentially used to belong to the electronic [100], security [101] or glass [102] industries.

Overall, 80% of articles used the extrusion-based printing technique. This trend was found to be highly consistent with the outcome of a recent systematic review dedicated to hydrogel bioprinting [23] that found extrusion to be used in 76% of 393 articles. It also agreed to a previous review over the privileged place of extrusion [103], in spite of a huge difference between the percentage of each printing technique involved (also first rank with 36.18% of 351 articles), in which the presence of cells within the biomaterial was mandatory in the search algorithm. Extrusion seems then to be by far the gold standard technique in non-involving cells’ strategies, which is consistent with the cells’ sensitivity to shear stress and associated loss of viability [104]. Moreover, due to the low viscosity nature of most materials investigated here and the inherent constraints of ink deposition this low viscosity triggered, extrusion was the most relevant tool to perform an efficient, reproducible printing process. Overall, a very limited number of studies investigated the association of cells [45,62,74] to the Si-derived material as part of the engineered final product to implant, and most of the publications relied on the host response rather than an enhanced biological activity due to additive cells. Only Wang et al. [74] found the addition of cells to their material to significantly improve bone formation.

In this review, the in vivo models investigated only concerned small animals. The rabbit appeared to be the most recurring target to assess bone repair, among other animal species or bone defects. This result goes against previous findings, as rat was reported by O’Loughlin et al. [105] (2008) to be the preferred model of fracture healing investigation. Some bone models possess analogous examples in humans or other species. Assessing various bones can therefore target a larger number of finer, physiologically equal models [106]. Hence, investigating the broadest range of animals and bones is crucial to provide an extensive and more truthful physiological bone response to implanted engineered biomaterials. However, in the effort to identify a baseline model to assess bone repair, thus providing a more standardized approach to enable comparison between studies and new materials, the diversity of animal and bone models might appear as a hurdle. The choice of a small animal model, beyond physiological fidelity, surgical accessibility or translation potential towards larger models, is associated with greater material advantages [107]. Indeed, smaller animals are more available and affordable in terms of basal caretaking costs *per capita* (purchase, housing, feeding, monitoring and overall costs of experiment-associated materials). Questioning here the bone regeneration potential of silica-based (bio)inks, the rabbit model appeared, hence, to be the most pragmatic and suitable model and logically federated the largest number of studies. Preclinical studies following the identification of the most outstanding materials regarding bone repair should consider the implantation in larger mammal models, as the studies presented here can only provide an in vivo first glimpse of a small model response.

Tightly bound to the animal species model choice, the baseline characteristics are also essential. Even though the outcome (a fully-filled, mature mineralized tissue with equal —if not superior—mechanical capacities regarding native tissue consecutively to creation of a bone defect) could be expected to be the major interest, the healing dynamics involved, the physiological performance and well-being of biological material over time should also be considered. In the scope of reaching the most reliable bone model to standardize the research of bone tissue engineering, extensive attention should then be brought with priority to the baseline characteristics of studied animals. First, the age of animals should be systematically reported, for this variable has a tight relationship with bone repair following a trauma [108,109]. Regarding gender, if fracture incidence was reported to be different through gender and age [110], there is no consensus in the literature regarding an influence of the sex over bone repair. Even though several studies in mice reported that males had a faster bone healing, they were unable to formally discriminate this result from the fact that studied males had more weight than females, or a superior activation of signaling pathways [111]. Overweight and obesity were indeed reported to delay and reduce bone healing post trauma [112,113].

To summarize, the age, weight and sex of animals should at least be reported systematically in each study. For example, as rabbit was the most abundant model in the articles scoped here (65% of total), it is proposed here to select specimens (either male of female) of 28–30 weeks of age for skeletal maturity [114,115] (specifically based on the growth plate closure age over life expectancy age), preferentially with homogeneous weights.

The choice of the defect location in animals is of major interest regarding healing dynamics through time. Each bone has its own indications for the creation of a defect [116], and studies may exhibit different approaches. Each model offers different relevant defects in terms of the physiological response and mechanical consequences, aiming or not for the creation of a critical-size defect [117]. For instance, flat bones were reported to heal same size defects faster than long bones [118]. Moreover, there seems to be an absence of consensus concerning the size of critical defects in humans [119], which adds challenge to the establishment of a universal defect for in vivo investigations. In this review, the question was to provide an extensive sight of the silica-derived inks regarding the enhancement of bone repair consecutive to the creation of a defect, beyond natural healing. Therefore, although of great importance, the size of the defect here was not determining, and this review focused on the ink formulation employed and the final outcome compared to appropriate controls. In this review, the majority of defects were associated with long bones, especially the femur (51% of articles). Regarding the most representative animal used, femoral defects created in rabbit varied from 5 × 5 mm to 6 × 10 mm cylindrical defects (33.3% of total articles). The most abundant defects (6 out of 17 (11.7% of total articles)) were associated with a 6 × 6 mm size [59,60,64,65,66,80]. Thus, in the challenge of proposing a consensual model regarding the development of silica-based inks for bone repair, the prevalence of 6 × 6 mm cylindrical femoral defects in rabbits in this review should be highlighted.

It was challenging for bone repair assessment to be quantitatively compared through the screened articles due to the heterogeneity in animal/bone defect/healing time models. First, the most recurrent techniques displayed to qualitatively appreciate and quantitatively measure bone regeneration were, respectively, histological staining and micro-computed tomography (micro-CT). Indeed, 78% (40 out of 51) of articles investigated bone repair with both histology and micro-CT. Most of the time, histological analysis remained qualitative, and the tissue organization and interaction with the implanted/remodeled material were described and compared to controls. In this regard, outcomes presented in this review were limited to the quality of bone repair in the presence of silica-derived materials.

Regarding articles’ quality and risk of bias, SYRCLE methodology highlighted a dramatic lack of animal procedure transparency and availability; ethical statements or animal procedures, as incorporated in studied articles, were unable to provide enough information to fill the required items. Most studies cited an ethical authority that provided an approval to conduct an in vivo experiment. However, until the full experimental procedure documents become available as a Appendix A attached to any animal experiment-related publication, the tools for risk of bias detection will widely remain irrelevant. We hereby emphasize the absence of animal experiment-related protocols attached to publications. A better transparency will improve publications at many levels, especially in terms of the reliability and reproducibility of conducted experiments. Unfortunately, likely due to potential patents and/or the intellectual property involved with publications, most of the animal procedure complete protocols remain unavailable, and therefore the displayed elements within most manuscripts cannot be considered as whole.

Concerning the regeneration process, the immune status of the animal model was questioned. It was considered that, in animals with an altered immune system, the conclusions of observed results could be inconsistent with a normal physiological response involving immune system key mechanisms [120,121]. Therefore, studies with immune-compromised animals were categorized as having an “Unclear” risk of bias.

Overall, although of great interest, the SYRCLE risk of bias assessment seemed not to represent the best tool to judge the quality of studies assessing the development of material formulations, for a majority of blinding-related aspects were conflictual with the implantation of a material versus an empty defect. ARRIVE 2.0 methodology was perceived as the opposite of SYRCLE regarding the proportion of items found to meet the requirement of the guidelines with the reported/unclear/not reported system. However, this scoring was not able to underline the weaknesses of some studies. For example, an inappropriate methodology employed regarding statistical hypotheses and tests could be performed in an article, or conflicts of interests could be displayed. According to the scoring system, the items *7-Statistical Methods* and *21-declaration of interests* would be considered reported and, consequently, provide a pledge to quality.

In spite of aiming for a single purpose, bone repair, this review highlighted the plurality in strategies: from the scaffold/ink preparation process to the delivery of the material. Most studies involved processes with several intermediary steps such as soaking, drying, sintering, etc., to generate a scaffold, while few were able to deliver a final product by one-step assembling and in situ printing [43,45]. Although the tissue engineering definition [122] depicted a trinity involving cells, adjuvants and support materials, almost none of the articles here investigated focused on a strategy relying on the use of cells, and most used an inorganic material-associated approach. This is consistent with a pragmatic will of obtaining a standardized, patient adjustable, quickly available elaborated material.

This manufacturing aspect raises the question of the materials’ availability for patients regarding the acceptable delay from the moment of demand versus the moment of delivery. Moreover, the questions about the strategy underlying the formulation are crucial, for the materials can be supposed to immediately replace a bone on a mechanical perspective with poor physiological respect, or to be associated with a longer (closer to natural) healing time but aim to restore the bone tissue as if it was native. Still regarding formulation, targeted physiological aspects can be crucial, choosing to stimulate vascularization, mineralization and innervation, or to first prevent a microbiological contamination hazard while keeping the composition as simple as possible, with a defined architecture and physicochemical properties expected to address physiologically-specific needs.

From a translation-to-clinic perspective, the manufacturing of new materials must therefore overcome considerable challenges. The scale up from the in vitro evaluation of materials with in vivo assessment in small animal models to in vivo therapeutic implantation to large mammals with respect to the laws and regulations for health seems yet to remain for a minority of products [123,124]. The increasing number and diversity of 3D printing approaches this last decade might help to bring more materials and applications to the clinic, and research should be continued in this effort.

## 5. Conclusions

Involved in bone tissue repair strategies since the 1970s, silicon and its derivatives recently grew in interest in the additive manufacturing and bioprinting fields. In this review, silicon appeared to be a versatile element to improve bone repair, whether as a constitutive or as a functionalizing, additive element. This review highlighted the dominant place of extrusion printing and the privileged place of rabbit and the femur, respectively, as animal and bone defect models for bone tissue engineering regarding the development of biocompatible inks suited for 3D printing. Considering the trend of an increasing number of publications related to the in vivo application of silica-derived inks for bone tissue engineering, it is expected that future printed patient-designed prostheses will integrate Si within the manufacturing process.

## Figures and Tables

**Figure 1 bioengineering-09-00388-f001:**
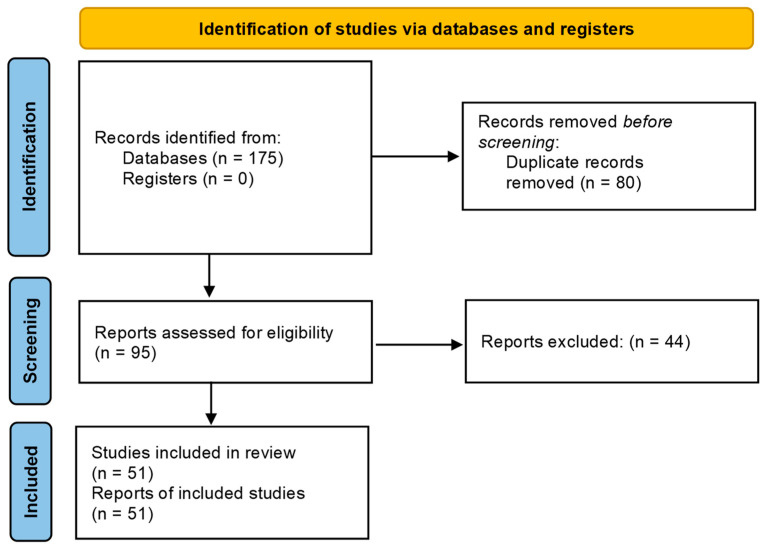
Flow diagram of study identification following PRISMA guidelines.

**Figure 2 bioengineering-09-00388-f002:**
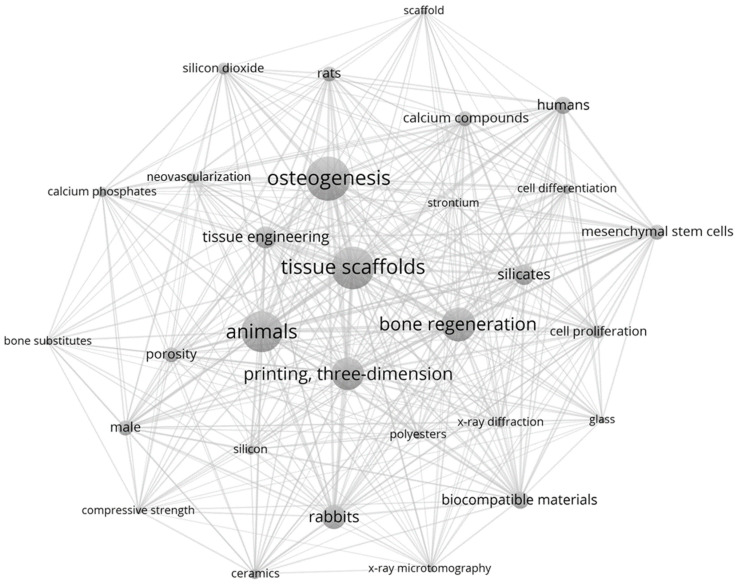
Science mapping analysis of scientific domains: keyword co-occurrence networks among the included articles. Each node size is proportionate to its degree and link’s thickness represents the tie strength.

**Figure 3 bioengineering-09-00388-f003:**
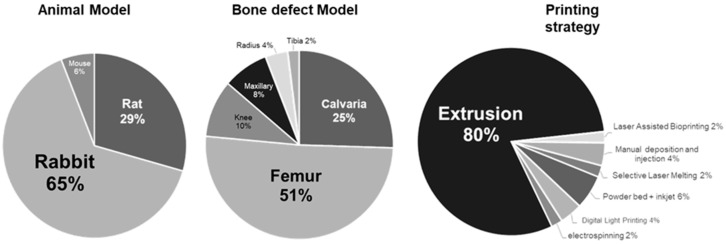
Overall animal/bone defect models and printing strategies repartition among the reviewed articles. Rabbit, femur and extrusion constituted the majority of the articles’ focuses.

**Figure 4 bioengineering-09-00388-f004:**
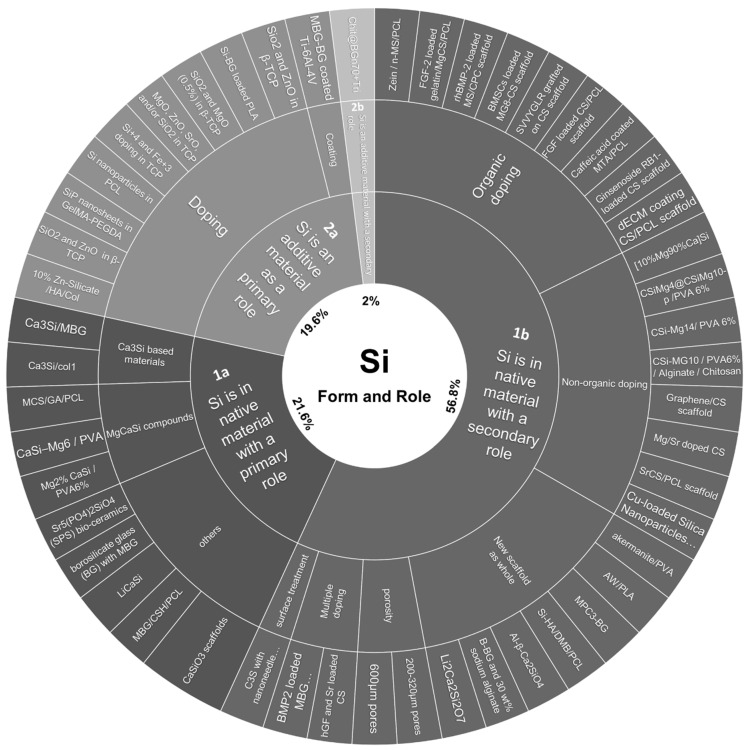
Overall silicon form and role repartition among the scoped articles.

**Figure 5 bioengineering-09-00388-f005:**
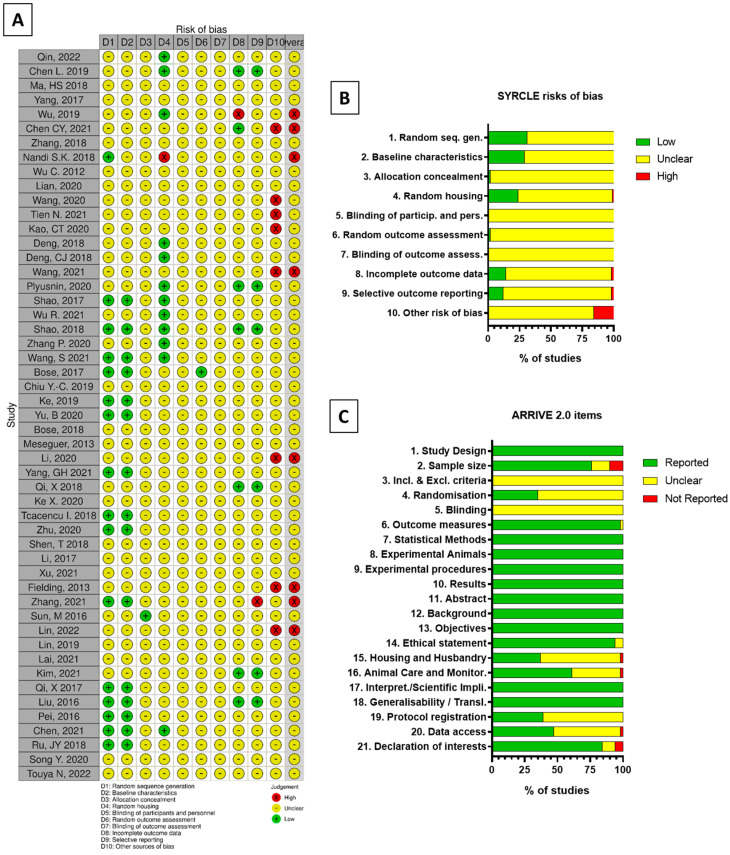
Visualizing risk of bias assessments of studied articles following (**A**,**B**) SYRCLE and (**C**) ARRIVE 2.0 guidelines with robvis tool.

**Table 1 bioengineering-09-00388-t001:** Systematic search strategy (PICO model).

Category	
Population	Animals with created bone defect
Intervention	Printed silica-based ink
Comparison	Untreated or controls
Outcome	Results of bone regeneration
Search combination	PubMed: ((silic*[Title/Abstract]) AND (print*[Title/Abstract]) AND (bone[Title/Abstract])) NOT (silicone[Title/Abstract]) AND ((in vivo[Title/Abstract]) Scopus: (TITLE-ABS-KEY (silic* AND print* AND bone AND NOT silicone) AND TITLE-ABS-KEY (in AND vivo)) Web of Science: **silic*** (Topic) AND **print*** (Topic) AND **bone** (Topic) NOT **silicone** (Topic) AND **in vivo** (Topic)
Language	English
Electronic databases	MEDLINE/PubMed, Scopus, Web Of Science

**Table 2 bioengineering-09-00388-t002:** Bone repair outcomes of the scoped articles sorted by silicon form, role and investigated formulation.

Si Form	Si Role	% of Articles	Main Focus	Number of Articles	Outcome	Publications
Si is in native material	with a primary role	21.6%	CaSiO_3_ scaffolds	2	- improved bone regeneration (vs. CaSiO_3_/Fe composite or pure Fe scaffold) - improved bone regeneration (vs. B-TCP)	Ma 2018 [47] Wu 2012 [48]
			MgCaSi compounds	3	- MCS/GA/PCL: 30% MCS formulation improved bone regeneration (vs. 0 and 15%) - CaSi–Mg6/PVA: Double layer CSi/PVA improved bone regeneration at 12 weeks (vs earlier times and vs. CaSi-Mg6/PVA single composites) - Mg2% CaSi/PVA6%: improved bone regeneration (vs. Ti scaffold)	Zhang 2018 [49] Shao 2017 [50] Wang 2021 [51]
			Sr_5_(PO_4_)_2_SiO_4_ (SPS) bio-ceramics	1	improved ICRS score (vs. CTR and TCP). Improved neobone structural organization	Deng 2018 [52]
			borosilicate glass (BG) with MBG	1	improved bone regeneration (vs. BG alone)	Qi 2018 [53]
			MBG/CSH/PCL	1	60% MBG: improved bone regeneration (vs. lower % BG or CSH/PCL alone)	Qi 2017 [54]
			C_3_S materials	2	- C_3_S/MBG: improved bone regeneration (vs. C_3_S alone and vs. ctrl) - C_3_S/Col1: no significant bone regeneration (vs. ctrl), or by adding cells (SCAPs)	Pei 2016 [55] Touya 2022 [45]
			LiCaSi	1	L_2_C_4_S: improved bone regeneration (vs. β-TCP)	Chen 2019 [56]
	with a secondary role (1/2)	56.8%	Organic doping	9	- dECM coating CS/PCL scaffold: improved bone regeneration (vs. CS/PCL alone) - Ginsenoside Rb1-loaded CS scaffold: improved bone regeneration (vs. CS alone) - Caffeic acid coated MTA/PCL: improved bone regeneration (vs. no coating) - FGF-loaded CS/PCL scaffold: improved bone regeneration (vs. CS/PCL alone) - SVVYGLR grafted on CS scaffold: improved bone regeneration (vs. CS alone) - BMSCs-loaded Mg8-CS scaffold: no significant difference in bone formation (vs. no BMSCs loading) - rhBMP-2-loaded MS/CPC scaffold: improved bone regeneration (vs. CS alone) - FGF-2-loaded gelatin/MgCS/PCL: improved bone regeneration (vs. gelatin/MgCS/PCL vs. MgCS/PCL) - Zein/n-MS/PCL: improved bone regeneration (20%ZN/30%n-MS/50%PCL vs. 10%ZN/30%n-MS/60%PCL vs. 0%ZN/30%n-MS/70%PCL)	Wu 2019 [57] Chen 2021 [58] Tien 2021 [59] Kao 2020 [60] Zhu 2020 [61] Shen 2018 [62] Li 2017 [63] Lai 2021 [64] Ru 2018 [65]
Si is in native material	with a secondary role (2/2)		Non-organic doping	8	- Cu-loaded silica nanoparticles in PLGA: improved bone regeneration (vs. SN alone or PLGA alone) - SrCS/PCL scaffold: improved bone regeneration (vs. CS/PCL alone) - Mg/Sr-doped CS: improved bone regeneration (vs. MgCS or SrCS) - Graphene/CS scaffold: improved bone regeneration (vs. 0 graphene) - Csi-Mg10/PVA6%/alginate/chitosan: improved bone regeneration (vs. Csi alone) - Csi-Mg14/PVA 6%: improved bone regeneration (vs. every inferior Csi-Mg compound) - CsiMg4@CsiMg10-p/PVA 6%: improved bone regeneration (vs. CsiMg4@CsiMg10 vs. CsiMg10@CsiMg4 vs. CsiMg10@CsiMg4-p) - (10%Mg90%Ca)Si: 8 weeks lower but 16 weeks improved bone regeneration (vs. bredigite and vs. pure CaSi)	Lian 2020 [44] Chiu 2019 [66] Lin 2022 [67] Lin 2019 [68] Ke 2020 [69] Sun 2016 [70] Chen 2021 [71] Shao 2018 [72]
			Multiple doping	2	- BMP2-loaded MBG-coated 1393BG: improved bone regeneration (vs. 1393 alone) (BMP2-loaded > MBG + 1393) - hGF and Sr-loaded CS: improved bone regeneration (vs. Sr loading alone and vs. CS alone)	Wang 2020 [73] Wang 2021 [74]
			New scaffold as whole	7	- LCS (Li_2_Ca_2_Si_2_O_7_): improved bone regeneration and ICRS (vs. ctrl and β-TCP) - B-BG and 30 wt% sodium alginate: improved bone regeneration (vs. HA and ctrl) - Al-β-Ca_2_SiO_4_: improved bone regeneration (15%Al vs. C2S alone) - Si-HA/DMB/PCL: qualitatively appreciated bone colonization - MPC3-BG: improved bone regeneration (vs. MPC alone and ctrl) - AW/PLA: improved bone regeneration (vs. AW alone or PLA alone) - akermanite (Ca(2)MgSi(2)O(7))/PVA: improved bone regeneration (vs. β-TCP)	Deng 2018 [75] Zhang 2020 [76] Yu 2020 [77] Meseguer 2013 [78] Li 2020 [43] Tcacencu 2018 [79] Liu 2016 [80]
			porosity	2	- 600 µm pores: most bone ingrowth, good mechanical strength and excellent Mg2+ ion release potential (vs. 480 and 720 µm pores) - 200–320 µm pores: slower degradability but improved bone regeneration (at 16 weeks vs. 450–600 µm pores)	Qin 2022 [41] Wu 2021 [40]
			surface treatment	1	C_3_S with nanoneedle surface structure: improved bone regeneration (vs. no treatment)	Yang 2017 [81]
Si is an additive material	with a primary role	19.6%	Doping	9	- SiO_2_ and ZnO in β-TCP scaffolds: improved bone regeneration (vs. no doping) - Si-BG-loaded PLA scaffolds: equal integration and bone regeneration (vs. Ti6Al4V and vs. empty defects) - SiO_2_ and MgO (0.5%) in β-TCP scaffolds: improved bone regeneration (vs. no doping) - MgO, ZnO, SrO and/or SiO_2_ in TCP: Mg/Si-doped scaffold improved bone regeneration (vs. all others and pure TCP) - Si + 4 and Fe + 3 doping in TCP: enhanced blood vessels and osteoid tissue formation (vs. TCP alone) - 100 nm Si particles in PCL: enhanced bone regeneration (vs. 500 and 800 nm particles) - SiP nanosheets in GelMA-PEGDA: enhanced angiogenic activity and osteogenic induction for bone regeneration (vs. BP/GelMA-PEGDA and GelMA-PEGDA alone) - SiO_2_ (0.5%) and ZnO (0.25%) in β-TCP scaffolds: improved bone regeneration (vs. no doping) - 10% Zn-Silicate/HA/Col: improved bone regeneration (vs. 10% Zn-Silicate/HA/Col + P38 inhibitor and 10% Zn-Silicate/HA…+ Col ERK1/2 inhibitor)	Nandi 2018 [39] Plyusnin 2020 [82] Bose 2017 [37] Ke 2019 [83] Bose 2018 [38] Yang 2021 [42] Xu 2021 [84] Fielding 2013 [85] Song 2020 [86]
			Coating	1	- MBG-BG-coated Ti-6Al-4V: improved bone regeneration (vs. no coating)	Zhang 2021 [46]
	with a secondary role	2%	Chitosan + BGn70 + Tri	1	Chit@BGn70 + Tri (tri-molecule dexamethasone-FGF2-phenamil-loaded chitosan/bioglass-coated PCL scaffold): improved bone regeneration (vs Chit@BGn70 vs. ctrl)	Kim 2021 [87]

**Table 3 bioengineering-09-00388-t003:** Decision table of articles sorting.

Category	Si Is in the Native Material	Si Is Primary in the Study
1a	Yes	Yes
1b	Yes	No
1c	No	Yes
1d	No	No

## Data Availability

All data gathered for this review were obtained from the scoped and cited articles. No new data were created or analyzed in this study.

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
