# Peer review of "In Vivo Application of Silica-Derived Inks for Bone Tissue Engineering: A 10-Year Systematic Review"

_bioengineering, 2022, doi:10.3390/bioengineering9080388_

Round 1
Reviewer 1 Report
This paper was well-organized, the silica-based inks play an important role for bone tissue engineering. I recommend the publication of this paper, some suggestions as follow:
1) in the Introduction section, some related papers related to 3D bioprinting could be cited, such as: https://doi.org/10.3390/cryst11040353
2) Are any papers related to bioinks, in Figure 2, "bioinks" was not appeared, however, bioinks in extrusion 3D bioprinting get more attentions in recent few years.
3) Some spelling mistakes appeared in the manuscript, please check the whole paper, also in Tables.
Reviewer 2 Report
The manuscript entitled "In vivo application of silica-derived inks for bone tissue engineering: a 10-years systematic review" is an interesting review of the scientific literature aimed at assessing the current use and outcomes of silica-derived inks to promote in vivo bone tissue regeneration.
The paper has several strengths, and this reviewer has particularly appreciated the use of PRISMA and PICO methods to perform this systematic review.
However, some points have to be addressed before publishing this work.
Introduction
Lines 32-33. Authors should support this sentence by adding an appropriate reference which focuses on the overall significance of musculoskeletal conditions worldwide, such as: Hernigou P, Scarlat MM. Growth in musculoskeletal pathology worldwide: the role of Société Internationale de Chirurgie Orthopédique et de Traumatologie and publications. Int Orthop. 2022 Jul 18. doi: 10.1007/s00264-022-05512-z.
After this part, authors should improve the section related to the strategies/products conventionally used by orthopaedists to promote bone healing.
When authors assert at lines 35-38 that “A wide range of synthetic biomaterials has been developed to overcome limitations associated with the use of autografts and allografts, including the risk of disease transmission, the morbidity on the donor site, the risk of infection and immunogenicity and the lack of availability” they should differentiate the advantages/disadvantages of autografts and allografts.
For instance, the risk of immune reactions is a potential issue related to the use of allogeneic bone tissues and not to autografts. Moreover, concerning the risk of possibility of infection, although this sentence is correct, it is worth noting that allografts are procured, processed, and distributed only by Tissue Banks, which operate under strict guidelines and sterile conditions-i.e., minimizing the abovementioned issues.
Therefore, this reviewer suggests adding this comment in the text.
To this aim, the following references may help the authors to better focus the field:
Vangsness CT et al. Overview of safety issues concerning the preparation and processing of soft-tissue allografts. Arthroscopy. 2006 Dec;22(12):1351-8. doi: 10.1016/j.arthro.2006.10.009.
Grassi FR et al. Design Techniques to Optimize the Scaffold Performance: Freeze-dried Bone Custom-made Allografts for Maxillary Alveolar Horizontal Ridge Augmentation. Materials (Basel). 2020 Mar 19;13(6):1393. doi: 10.3390/ma13061393.
In addition to that, the use of xenografts should be briefly mentioned as an alternative treatment option. In this respect, please note that several Companies have in their portfolio commercial products containing bone tissue from different animal sources. Please read the following recent paper to have an overview about commercial allografts and xenografts commonly used by clinicians as current alternatives to autografts:
Govoni M et al. Commercial Bone Grafts Claimed as an Alternative to Autografts: Current Trends for Clinical Applications in Orthopaedics. Materials (Basel). 2021 Jun 14;14(12):3290. doi: 10.3390/ma14123290.
Results
Line 254. The legend of Table 2 is missing.
Table 2. Please check the vertical text of the first 2 columns because there are some typos.
Discussion
Since the focus of this review is the in vivo application of silica-derived inks for bone tissue engineering, this reviewer recommends adding a new summary table containing information on the animal model characteristics, such as species, sex, age, weight, defect location, in accordance with what is reported in the Discussion section at lines 416-478.
On the other hand, the author could consider enriching the existing Table 2 with the abovementioned information.
Reviewer 3 Report
This paper has reviewed recent research progresses In vivo application of silica-derived inks for bone tissue engineering. The paper is structured with clear, emerging The inks and materials are reviewed. Overall, the paper is a good round up overview on the scope of the current research. As far as reviewer's knowledge, several comments on the current draft is listed at below:
1. The review is basically done in a reciting way, which lacks in-depth discussion on the substantial merits or disadvantages of certain classification method, etc. Furthermore, the review lacks comparison across literatures, but only lists the results one by one.
2. Although the review is clearly structured in several different perspectives, the discussion in one section is really not focusing on the specific topic, but aliased with other aspects..
3. The resolution of some figures are too low.
4. There are so many paragraphs in one section. It looks very redundant.
Round 2
Reviewer 2 Report
I have no further comments.
Therefore, I recommend this paper for publication.